# Knowledge and practices surrounding malaria and LLIN use among Arab, Dazagada and Fulani pastoral nomads in Chad

**Azoukalné Moukénet** [1,2]*, **Sol Richardson** [3,4], **Kebféné Moundiné**[5], **Jean Laoukolé**[6], **Ngarkodje Ngarasta**[2], **Ibrahima Seck**[1]

**1** Cheikh Anta Diop University, Dakar, Senegal, **2** University of Ndjamena, Ndjamena, Chad, **3** Malaria Consortium, London, United Kingdom, **4** Vanke School of Public Health, Tsinghua University, Beijing, China, **5** StraDEC Training, Research & Innovation Department, Ndjamena, Chad, **6** National Malaria Control Program, Ndjamena, Chad

* azoukalne07@yahoo.fr

## Abstract

### Background

Chadian pastoral nomads are highly exposed to malaria due to their lifestyle and their mobility between various endemic areas. To inform strategies to reduce nomads' risk of malaria and associated morbidity and mortality, it is important to understand the factors associated to their knowledge of malaria transmission and prevention practices.

### Methods

A cross–sectional study among Arab, Dazagada and Fulani pastoral nomadic groups was conducted in February and October 2021. A validated structured electronic questionnaire was administered to assess knowledge of malaria. Attitudes and malaria prevention practices were assessed on the basis of perception of the causes of malaria and the use of a long-lasting insecticide-treated net (LLIN) the day before the survey. Data were analyzed using Chi—square tests and multivariate logistic regression with covariates adjustment.

### Results

A total of 278 nomads aged 20 to 65 years were included in the study. Overall, 90.7% of participants surveyed had a good knowledge of malaria. Fulani respondents were more likely to have a good knowledge of malaria than Arab respondents (Adjusted Odd ratio (AOR): 5.00, 95% CI: 1.04–24.03) and households possessing a LLIN were more likely to have a good knowledge of malaria (AOR: 9.66, 95% CI: 1.24–75.36). Most nomad households surveyed reported sleeping under a mosquito net the night before the survey (87.1%) while 98.9% owned a LLIN. Daza respondents (AOR: 0.23, 95% CI: 0.09–0.56) were less likely to use LLINs than Arab respondents. The middle (AOR: 2.78, 95% CI: 1.17–6.62) and wealthier households (AOR: 6.68, 95% CI: 3.19–14.01) were more likely to use LLINs. Knowledge of malaria was associated with the use of LLIN (AOR: 12.77, 95% CI: 1.58–102.99).

**Data Availability Statement:** All relevant data are within the paper and its Supporting information files.

**Funding:** The author(s) received no specific funding for this work.

**Competing interests:** The authors have declared that no competing interests exist.

**Abbreviations:** ANC, Antenatal Care; AOR, Adjusted odd ratio; COR, Crude odd ratio; IPTp, Intermittent Preventive Treatment (pregnant women); LLIN, Long–lasting insecticide–treated net; NMCP (PNLP), National Malaria Control Program (*Programme National de Lutte contre le Paludisme*); PCA, Principal Component Analysis; SMC, Seasonal malaria chemoprevention; SPAQ, Sulphadoxine-Pyrimethamine and Amodiaquine; StraDEC, Strategic Decisional Engineering Consulting sarl; WHO, World Health Organization.

## Conclusion

There remains a need to improve nomads' understanding of Plasmodium falciparum-carrying mosquitoes as the vector for malaria transmission and the quality of information provided. Knowledge of malaria and its prevention strategies in nomadic setting lead to the use of LLINs. Further reductions in malaria morbidity can be achieved by improving nomads' access to LLINs. This study can inform on the design policies to improve nomadic communities' knowledge of malaria prevention and promoting LLIN use as requested by the national policy against malaria.

## Introduction

In Chad, malaria is endemic across most regions with areas at risk of epidemics. It remains the main cause of consultation at health facilities and hospitalizations [1]. In 2017, one study found the malaria prevalence of 7.7% in general population and 8.8% in children aged 59 months in the Sahelo–Saharian region of Chad [2]. Another study found in the same region a malaria prevalence up to 30% among the nomadic population [3]. Due to their lifestyle, nomadic populations are highly exposed to malaria [4]. In addition, because of their living environment (near settlements or on the islands of Lake Chad) [5], their temporary area of living in the south of the country [6] and their lack of immunity to malaria [7, 8] nomads are much more exposed to malaria than the settle population.

To tackle malaria dynamic by reducing its related morbidity and mortality, a number of interventions including vector control, chemoprevention, and improved access to diagnostic tests and treatment, have been implemented as part of the national policy against malaria [9]. With the aim to reduce barriers to health interventions, the policy insists on free-of-charge access to essential interventions against malaria including prevention, diagnostic, case management and communication for behavior change. The policy provide guidance for developing a specific health strategy for isolate population including nomads. Thus, the National Health Program in charge of nomads advocated for improving access of nomads to health interventions [10].

The national policy against malaria has highlighted that all malaria interventions should include communication for behavior change related to the good and regular use of long-lasting insecticide-treated nets (LLINs), adherence to indoor residual spray, Seasonal malaria chemoprevention (SMC), the attendance at antenatal consultation (ANC), Intermittent Preventive Treatment (IPTp), and the early attendance at health facilities in case of first symptoms of malaria.

Among malaria prevention strategies adopted in Chad, intermittent preventive treatment has been shown to be effective in preventing malaria among pregnant women elsewhere [11]. In addition, SMC with sulphadoxine-pyrimethamine and amodiaquine (SPAQ) has been proven to be effective in preventing seasonal malaria among children aged 3–59 months in the Sahel region of Africa [12–19]. SMC programmes have been rolled out, targeting children aged 3–59 month across the Sahel region of Chad. IPTp is also administered to pregnant women from the fourth month of pregnancy during their antenatal care in health facilities. The distribution of LLINs has proven to be most promising [20], and highly cost-effective in preventing malaria cases among children [21] and pregnant women [22]. In Chad, LLINs are primarily distributed through mass campaigns every three years, and routinely at health facilities for pregnant women during their first ANC and for children aged 0–11 months during routine vaccination [9]. The last mass LLIN distribution campaign in Chad occurred in 2020.

In nomadic settings, children often receive SMC only when they are living temporarily in the Sahel region during SMC implementation. Sometimes, they are excluded from SMC as they are often uncounted during census enumeration [23]. Regarding IPTp, nomadic women face geographical barriers to accessing antenatal clinics, or feel excluded from services available in the locations where they take up temporary residence [6, 24]. Concerning LLINs, mass distribution campaigns have no strategy for enumeration of nomadic populations. In addition, nomads may not receive LLINs during routine distributions since they frequently travel in remote areas; they may have infrequent contact with the health system, sometimes only when an illness has progressed to an advanced or serious condition [25]. However, anecdotal suggest that most nomads purchase mosquito nets themselves which indicates a potential for its use [26].

Though, ownership of LLINs does not guarantee their use due to behavioral factors [27]. Evidence shows that, LLIN use is driven by users' knowledge, the household's economic status, the size of household [27, 28], the rainy season and presence of mosquitoes [29–31]. The identification of factors which predict demand for mosquito net among nomads would allow the National Malaria Control Program (NMCP) to tailor interventions to control malaria among this group and meet their specific needs.

In Chad, nomads represent 3.5% of the population [32]. They live mainly between the Saharan and Sahelian zones of the country. Nomads move with their flocks to find pasture and water; although they have more permanent dwellings in the north, spend dry seasons in the comparatively wetter south and, return north at the beginning of the rainy season (mid-April–May). A large number of pastoralists move with their herds from southern regions with high malaria transmission zones toward the provinces of Chari Baguirmi and Hadjer Lamis. Following climatic [33], economic [34] and political [35] changes over the past decades, a considerable increase in pastoral mobility has been recorded in the Sudanian zone [36].

Theories of health behavior change show that the knowledge about the causation, transmission, prevention and treatment of malaria may facilitate changes in attitude, resulting in the adoption of positive preventive practices that can reduce the risk of exposure to malaria [37]. Thus, understanding local factors affecting the perception of the causes and modes of transmission of malaria [38], and prevention practices [39–45], is key to inform strategies to reduce nomads' risk of malaria and associated morbidity and mortality. However, previous studies on malaria in nomadic communities in Chad have mostly focused on the epidemiology of the disease, and very limited evidence is available on the level of knowledge of malaria and its associated factors. In addition, little is known about the coverage and use of mosquito nets. In order to provide evidence to inform efforts by the Chadian Ministry of Health and partners to tailor interventions for nomadic communities, this study aimed to identify factors associated to the knowledge about malaria and LLIN practice among the main nomadic groups in Chad. Our motivation for conducting this study at this time was to inform policy decisions to tailor interventions to control malaria among this group and meet their specific needs relating to LLINs.

## Methods

### Population and study site

The study was carried out in the provinces of Hadjer Lamis and Chari Baguirmi in the Sahelian region of Chad. According to the national census, nomads represent 3.7% and 4.0% of the population of Hadjer Lamis and Chari Baguirmi respectively [32]. In comparison to the national level (40.9%), the prevalence of malaria is moderate in Hadjer Lamis (15.9%) and high in Chari Baguirmi (37.2%) [2]. SMC is being implemented in both of these provinces targeting children aged 3–59 months, in addition to IPTp, and routine and mass LLIN distribution

campaign. In these provinces, the percentage of population owning at least one mosquito net are 76.3% (71.7% for LLINs) in Hadjer Lamis and 94.5% (74.1% for LLINs) in Chari Baguirmi [2]. The use of mosquito net is low in Hadjer Lamis (12.6% for ordinary nets and 12.4% for LLINs), but relatively high in Chari Baguirmi (67.6% for ordinary nets and 43.1% for LLINs) [46].

The three main nomadic groups, the Arabs, Dazagada and Fulani, were included in the study as they travel through a wide range of malaria transmission settings, and findings from these groups can be generalizable to other nomadic groups. The three groups represent 91.5% of nomads population in Chad, of which 45.8% of nomads are Dazagada/Gorane, 38.4% are Arabs and 7% are Fulani [32]. To capture experiences of nomads regarding malaria, a questionnaire was administered to men or women aged above 18 years from selected households who provided consent to participate in the study.

## Study design

During 8–20 February 2021 and 14–18 October 2021 a cross sectional survey was conducted among nomads in Dourbali and Massenya districts in Chari Baguirmi province and Massaguet district in Hadjer Lamis province.

## Sampling

Within each nomad group a multi-stage cluster random sampling technique with the first-stage the camp and the second-stage the household was used and lead to a minimum sample of 270 study subjects. At the first level, from the list of camps for each three nomad groups provide by its leaders we randomly selected 135 camps using a random number draw. At the second stage, within each camp selected, surveyors used random number draws to select two households. For household selected, one member of household older than 18 years was requested for interviews and responded on behalf of the household to which he or she belongs.

## Data collection method

With a view to measure the level of knowledge of malaria and the use of preventive methods by nomads, a structured electronic questionnaire was developed based on Peto et al. [47]. The questionnaire was implemented in KoBoCollect v2021.2.4 [48], and was administered offline with responses uploaded to the server once WiFi connection was available. The survey questionnaire was administered in February and October 2021 by three trained data collectors fluent in the local languages and used to collect data for nomad immunization programs. The questionnaire comprised items on respondents' socio-demographic characteristics; their knowledge and experiences regarding malaria; and their use of mosquito nets. The survey was carried out 1–2 months after the 2020 mass LLIN distribution campaign.

## Selection of variables

We assessed respondents' knowledge about LLINs, causes and symptoms of malaria, and malaria prevention practices. The questionnaire included a number of questions for this purpose, listed below with response categories:

• Knowledge about LLINs: 1) Sleeping under a LLIN as means to protect against malaria, 2) Age groups targeted by LLIN routinely distributed at health facilities, 3) When to go under a mosquito net.

- Coverage of LLINs: 1) Own at least one mosquito net, 2) Own at least one LLIN, 3) Number of mosquito net owned, 4) Type of mosquito net owned.

- Practices surrounding LLINs: 1) Slept under mosquito net the night before the survey.

- Knowledge about malaria risks: 1) Period of high incidence (June, July–September, October), 2) Groups most at risk (children, pregnant women, adults, persons with disabilities).

- Knowledge about malaria causes: 1) Mosquito bite, 2) Environmental cause of malaria (water/rain, hot/sun), 3) Poor nutrition (hunger, food), 4) Religious/supernatural causes (destiny/fate/act of God, magic/witchcraft).

- Knowledge about common symptoms of malaria: 1) Fever, 2) Chills, 3) Muscle pain, 4) Stomach pain, 5) Diarrhea, 6) Nausea, 7) Vomiting.

- Respondents' socio-demographic characteristics, including age, gender of the head of household, marital status, household size, and socioeconomic status calculated from wealth index (S1 File).

## Determination of knowledge and practices regarding malaria and LLINs

**Knowledge of malaria.** Common principles used to measure knowledge of malaria include questions on transmission and preventive interventions [49]. This study used similar principles, with questions on dimensions of transmission related periods of high transmission (rainy season), the group most at risk (children and pregnant women), means of transmission (mosquito bite) and common symptoms (fever, chills, muscle pain, stomach pain, diarrhea, nausea and vomiting). The dimension of interventions related to sleeping under a LLIN as mean of protection against malaria. Each correct response to question was scored one point and zero for wrong answers. An overall knowledge score was calculated by summing the scores for each respondent across all questions. Those with scores of 2.5 (mid-point between 0–5) or above were considered to have good knowledge, while those with lower scores were categorized as having poor knowledge about malaria.

**Practices regarding malaria and LLINs.** Good malaria prevention practices were assessed based on the ownership and use of LLINs at night. Respondents were considered to have good prevention practices if they declared that they both owned and slept under a LLIN the night before the survey. Poor practice was defined as anyone of the following: not sleeping under a LLIN, or sleeping under an ordinary (non-treated) mosquito net.

## Data analysis

Principal Component Analysis (PCA) [50] was used to develop wealth categories for the households based on access to facility including potable water and ownership of durable assets including solar kit, radio, telephone, cart tracked by animal, motorcycle/scooter, and caws/camels and sheep/goats per capita. Access and ownership was coded as 0 or 1 and missing cases were excluded. The first dimension of the PCA was taken as the household wealth score and range into tertiles; households were then placed into socioeconomic categories based on their scores.

Regarding univariate and multivariate analysis, we first performed a descriptive analysis and presented participants' social and demographic characteristics stratified by nomadic groups (Arab/Dazagada/Fulani). We then employed Chi–square tests to assess any significant difference in knowledge of prevention, causes, symptoms, and practices between nomad

groups. We then conducted logistic regression analysis to identify the factors associated with knowledge and practice of malaria among nomads in Chad. Crude and adjusted odds ratios (OR) were calculated to check statistical associations between the dependent and independent variables using the binary logistic regression and multivariable logistic regression models. All variables in the study were initially tested for association with good knowledge and practice regarding malaria and LLINs using a binary logistic regression model. Those which showed a significant statistical association (p < 0.05) were added to the multivariable analysis model to assess whether the association existed after controlling against all the rest of the variables. A 95% confidence intervals and the 5% significance level were calculated for all odds ratios. Data analyses were conducted using Stata 13.

### Ethics approval and consent to participate

Oral consent was obtained from participants prior to the study. The study was approved by the National Bioethics Committee of Chad (N˚ 0193/PR/MESRI/SG/CNBT/2020, September 21, 2020).

### Inclusivity in global research

Additional information regarding the ethical, cultural, and scientific considerations specific to inclusivity in global research is included in the S4 File.

## Results

### Sample characteristics

A total of 278 surveyed participants aged 20–65 years were included in the study. Basic socio-demographic characteristics of the participants are presented as frequencies and percentages in Table 1. Most participants were male (68.0%), married (92.4%) and, aged above 40 years

**Table 1. Socio demographic characteristics of survey participants by nomadic group, frequency (%).**

| Variables | Category | Arab (n = 105) | Daza (n = 84) | Fulani (n = 89) | Chi 2 Statistic | p–value (for difference between groups) | All (n = 278) |
|---|---|---|---|---|---|---|---|
| **Gender** | Female | 35 (33.3) | 29 (34.5) | 25 (28.1) | 1.0 | 0.620 | 89 (32.0) |
| | Male | 70 (66.7) | 55 (65.5) | 64 (71.9) | | | 189 (68.0) |
| **Marital status** | Widowed/divorced | 8 (7.6) | 6 (7.1) | 7 (7.9) | 2.3 | 0.683 | 21 (7.6) |
| | Monogamy | 74 (70.5) | 66 (78.6) | 68 (76.4) | | | 208 (74.8) |
| | Polygamy | 23 (21.9) | 12 (14.3) | 14 (15.7) | | | 49 (17.6) |
| **Age** | 20–29 | 15 (14.3) | 0 (0.00) | 14 (15.7) | 17.3 | 0.002 | 29 (10.4) |
| | 30–39 | 29 (27.6) | 29 (34.5) | 34 (38.2) | | | 92 (33.1) |
| | ≥40 | 61 (58.1) | 55 (65.5) | 41 (46.1) | | | 157 (56.5) |
| **Household size** | 2–5 | 29 (27.6) | 22 (26.2) | 26 (29.2) | 4.2 | 0.654 | 77 (27.7) |
| | 6–8 | 40 (38.1) | 40 (47.6) | 41 (46.1) | | | 121 (43.5) |
| | 9–11 | 26 (24.8) | 15 (17.9) | 13 (14.6) | | | 54 (19.4) |
| | ≥12 | 10 (9.5) | 7 (8.3) | 9 (10.1) | | | 26 (9.4) |
| **Number of children under 5 years** | 1–2 | 82 (78.1) | 50 (59.5) | 68 (76.4) | 9.3 | 0.010 | 200 (71.9) |
| | ≥3 | 23 (21.9) | 34 (40.5) | 21 (23.6) | | | 78 (28.1) |
| **Economic status** | Poor | 34 (32.4) | 46 (54.8) | 32 (36.0) | 19.7 | 0.001 | 112 (40.3) |
| | Middle | 15 (14.3) | 19 (22.6) | 20 (22.5) | | | 54 (19.4) |
| | Rich | 56 (53.3) | 19 (22.6) | 37 (41.5) | | | 112 (40.3) |

(56.5%), with age distributions varying significantly by nomadic group (Table 1). A majority of the households had more than six members (92.3%) and less than three children under five years old (71.9%). The number of children under five years, the marital and economic status varied significantly by nomadic group with a higher proportion of Daza being categorized poor than other groups.

## Malaria–related knowledge and associated factors

A majority of the respondents (77.0%) reported that malaria occurs mostly during rainy season July–September. Contrary to the Daza group, Arab and Fulani respondents reported that malaria is also more frequent at the beginning and end of the rainy season spanning June–October. The period of high transmission of malaria reported by participants were significantly dependent on nomadic group. In Table 2, nomad groups reported that groups most at risk of malaria are children under five years (86.0% of respondents) and pregnant women (73.0%). The groups most at risk of malaria reported varied significantly by nomad groups (Table 2).

Regarding causes of malaria, 68.3% of respondents correctly reported that mosquito bites could cause malaria. While a high proportion accepted that environmental factors such as water (61.5% of respondents) and heat/sun (31.7%) can be the main cause of malaria. Other nomads surveyed reported that poor nourishment (hunger for 42.4%, food for 38.5%), and a low proportion of respondents mentioned religious or supernatural forces (4.7% for destiny/ fate/act of God and 7.2% for magic/witchcraft) as the primary cause of malaria. Mentions of environment and poor nourishment as cause of malaria varied significantly by nomad groups.

Moreover, a majority of respondents were able to identify the main symptoms of malaria. Symptoms of malaria reported varied significantly by nomad ethnic groups (Table 2).

Of the socio demographic characteristics of nomads surveyed, ethnic group (OR: 4.58, 95% CI: 0.98–21.48) and possession of a LLIN (OR: 13.75, 95% CI: 1.83–103.20) were significantly associated with knowledge of malaria. Fulani respondents were five times more likely to have a good knowledge of malaria in comparison to Arab respondents, while households possessing a LLIN were 14 times more likely to have a good knowledge of malaria than households without LLINs (Table 3).

After adjusting for others socio-demographic characteristics, ethnic group and the possession of a LLIN were significantly associated with the good knowledge of malaria (Table 3). In comparison with Arab respondents, Fulani respondents were five times more likely to have a good knowledge of malaria (AOR: 5.00, 95% CI: 1.04–24.03). In addition, after adjustment for socio-demographic variables, ownership of a LLIN was independently associated with malaria knowledge. Households with a LLIN were 10 times more likely to have a good knowledge of malaria than households without LLINs (AOR: 9.66, 95% CI: 1.24–75.36).

## Mosquito net knowledge, ownership and use

Regarding knowledge of mosquito nets, 79.9% of respondents were aware of mosquito net as means to prevent mosquito bites. Furthermore, 91.4% of nomadic households surveyed owned at least one mosquito net, with 32.0% owning at least one LLIN. However, 87.1% of respondents (95.3% of respondents owning a mosquito net) reported sleeping under a mosquito net the night before the survey. Around half of respondents (54.3%) reported going under a mosquito net late from 19:00 or after. The ownership and use of mosquito net varied significantly by nomad ethnic groups (Table 2).

The socio demographic characteristics of nomads surveyed such as ethnic group, the head of household's marital status, the number of household members and that of children under

**Table 2. Reported knowledge of malaria and practice by nomadic group, frequency (%).**

| Variables | Category | Arab (n = 105) | Daza (n = 84) | Fulani (n = 89) | Chi 2 Statistic | p–value (for difference between groups) | All (n = 278) |
|---|---|---|---|---|---|---|---|
| **Reported period of highest malaria transmission** | June | 56 (53.3) | 8 (9.5) | 45 (50.6) | 44.7 | < 0.001 | 109 (39.2) |
| | July–September | 83 (79.0) | 71 (84.5) | 60 (67.4) | 7.5 | 0.023 | 214 (77.0) |
| | October | 55 (52.4) | 33 (39.3) | 47 (52.8) | 4.1 | 0.126 | 135 (48.6) |
| **Group most at risk of malaria** | Children under 5 years | 97 (92.4) | 59 (70.2) | 83 (93.3) | 24.7 | < 0.001 | 239 (86.0) |
| | Pregnant Women | 74 (70.5) | 55 (65.5) | 74 (83.1) | 7.4 | 0.025 | 203 (73.0) |
| | Adult | 50 (47.6) | 19 (22.6) | 59 (66.3) | 33.3 | < 0.001 | 128 (46.0) |
| | Disabled | 18 (17.1) | 19 (22.6) | 13 (14.6) | 2.0 | 0.375 | 50 (18.0) |
| **Vector cause of malaria** | Mosquito | 75 (71.4) | 52 (61.9) | 63 (70.8) | 2.3 | 0.314 | 190 (68.3) |
| **Environmental cause of malaria** | Water | 73 (69.5) | 52 (61.9) | 46 (51.7) | 6.5 | 0.039 | 171 (61.5) |
| | Heat/Sun | 28 (26.7) | 39 (46.4) | 21 (23.6) | 12.4 | 0.002 | 88 (31.7) |
| **Poor nourishment cause of malaria** | Hunger | 34 (32.4) | 45 (53.6) | 39 (43.8) | 8.7 | 0.013 | 118 (42.4) |
| | Food | 47 (44.8) | 33 (39.3) | 27 (30.3) | 4.3 | 0.118 | 107 (38.5) |
| **Religious/supernatural forces cause of malaria** | Destiny/fate/act of God | 7 (6.7) | 0 (0.0) | 6 (6.7) | 5.9 | 0.052 | 13 (4.7) |
| | Magic/witchcraft | 9 (8.6) | 3 (3.6) | 8 (9.0) | 2.4 | 0.304 | 20 (7.2) |
| **Sign/symptom malaria** | Fever | 89 (84.8) | 54 (64.3) | 84 (94.4) | 27.2 | < 0.001 | 227 (81.7) |
| | Chills | 38 (36.2) | 9 (10.7) | 23 (25.8) | 16.1 | < 0.001 | 70 (25.2) |
| | Muscle pain | 46 (43.8) | 62 (73.8) | 33 (37.1) | 26.5 | < 0.001 | 141 (50.7) |
| | Stomach pain | 26 (24.8) | 20 (23.8) | 19 (21.3) | 0.3 | 0.850 | 65 (23.4) |
| | Diarrhea | 26 (24.8) | 17 (20.2) | 26 (29.2) | 1.9 | 0.393 | 69 (24.8) |
| | Nausea | 18 (17.1) | 6 (7.1) | 13 (14.6) | 4.2 | 0.120 | 37 (13.3) |
| | Vomit | 88 (83.8) | 50 (59.5) | 71 (79.8) | 16.2 | < 0.001 | 209 (75.2) |
| **LLIN mentioned as preventive methods** | | 79 (75.2) | 62 (73.8) | 81 (91.0) | 10.2 | 0.006 | 222 (79.9) |
| **SMC mentioned as preventive methods** | | 13 (12.4) | 1 (1.2) | 8 (9.0) | 8.2 | 0.016 | 22 (7.9) |
| **ITPp mentioned as preventive methods** | | 8 (7.6) | 0 (0.0) | 8 (9.0) | 7.5 | 0.023 | 16 (5.8) |
| **Go under mosquito net from 7 PM** | | 55 (52.4) | 49 (58.3) | 47 (52.8) | 0.8 | 0.675 | 151 (54.3) |
| **Own at least one mosquito net** | | 105 (100.0) | 60 (71.4) | 89 (100.0) | 60.7 | < 0.001 | 254 (91.4) |
| **Own at least one LLIN** | | 43 (41.0) | 9 (10.7) | 37 (41.6) | 23.2 | < 0.001 | 89 (32.0) |
| **Mosquito net installed** | | 25 (23.8) | 14 (16.7) | 89 (28.1) | 3.2 | 0.198 | 64 (23.0) |
| **Last night slept under mosquito net** | | 103 (98.1) | 50 (59.5) | 89 (100.0) | 84.7 | < 0.001 | 242 (87.1) |
| **Received visit for SMC** | | 34 (32.4) | 2 (2.4) | 27 (30.3) | 28.4 | < 0.001 | 63 (22.7) |
| **Received at least one dose IPTp during last pregnancy*** | | 12 (32.4) | 2 (6.5) | 7 (25.9) | 6.9 | 0.031 | 21 (22.1) |

Note:

*Arab (n = 37), Daza (n = 31) and Fulani (n = 27); Percentage total exceed 100 because of multiple responses.

five years, the knowledge of malaria and the economic status of household were associated to the use of LLINs (Table 4). Daza respondents (OR: 0.16, 95% CI: 0.07–0.37) were one-sixth times more likely to use LLINs compared with Arabs. Monogamous households (OR: 0.37, 95% CI: 0.20–0.72) were around one-third times more likely to use LLIN in comparison to household lead by a widowed or divorced. Large households were more likely to use LLINs; in comparison to households with less than six members those with 9–11 members were around twice as likely to use LLINs (OR: 2.01, 95% CI: 0.93–4.35), and those with over 12 members were around six times more likely to use LLINs (OR: 6.07, 95% CI: 2.28–16.15). Households with more than two children under five years were twice as likely to use LLINs in comparison to households with 1–2 children under five years (OR: 2.05, 95% CI: 1.18–3.55). Respondents

**Table 3. Factors associated with knowledge of malaria among nomads.**

| Variables | Categories | Frequency (%) | Poor n = 26 | Accurate n = 252 | COR (95% CI) | p–value | AOR (95% CI) | p–value |
|---|---|---|---|---|---|---|---|---|
| **Ethnic group** | Arab (ref) | 105 (37.8) | 10 (9.5) | 95 (90.5) | 1 | | 1 | |
| | Daza | 84 (30.2) | 14 (16.7) | 70 (83.3) | 0.53 (0.22–1.25) | 0.147 | 0.69 (0.28–1.71) | 0.418 |
| | Fulani | 89 (32.0) | 2 (2.2) | 87 (97.8) | 4.58 (0.98–21.48) | 0.054 | 5.00 (1.04–24.03) | 0.044 |
| **Size of household** | 2–5 (ref) | 77 (27.7) | 10 (13.0) | 67 (87.0) | 1 | | 1 | |
| | 6–8 | 121 (43.5) | 13 (10.7) | 108 (89.3) | 1.24 (0.51–2.99) | 0.632 | 1.41 (0.55–3.56) | 0.473 |
| | 9–11 | 54 (19.4) | 2 (3.7) | 52 (96.3) | 3.88 (0.81–18.48) | 0.089 | 4.28 (0.86–21.19) | 0.075 |
| | ≥12 | 26 (9.4) | 1 (3.8) | 25 96.2) | 3.73 (0.45–30.66) | 0.220 | 2.29 (0.25–21.09) | 0.463 |
| **Owned a LLIN** | No (ref) | 185 (67.5) | 25 (13.5) | 160 (86.5) | 1 | | 1 | |
| | Yes | 89 (32.5) | 1 (1.1) | 88 (98.9) | 13.75 (1.83–103.20) | 0.011 | 9.66 (1.24–75.36) | 0.030 |
| **Gender** | Female (ref) | 89 (32.0) | 9 (10.1) | 80 (89.9) | 1 | | NA | NA |
| | Male | 189 (68.0) | 17 (9.0) | 172 (91.0) | 1.14 (0.49–2.66) | 0.765 | | |
| **Head of household status** | Widowed/ divorced (ref) | 21 (7.6) | 3 (14.3) | 18 (85.7) | 1 | | NA | NA |
| | Married monogamy | 208 (74.8) | 21 (10.1) | 187 (89.9) | 1.48 (0.40–5.46) | 0.553 | | |
| | Married polygamy | 49 (17.6) | 2 (4.1) | 47 (95.9) | 3.92 (0.60–25.41) | 0.152 | | |
| **Number children under 5 years** | 1–2 (ref) | 200 (71.9) | 19 (9.5) | 181 (90.5) | 1 | | NA | NA |
| | ≥3 | 78 (28.1) | 7 (9.0) | 71 (91.0) | 1.06 (0.43–2.64) | 0.892 | | |
| **Wealth categories** | Poorest (ref) | 112 (40.3) | 8 (7.1) | 104 (92.9) | 1 | | NA | NA |
| | Middle | 54 (19.4) | 5 (9.3) | 49 (90.7) | 0.75 (0.23–2.42) | 0.635 | | |
| | Richest | 112 (40.3) | 13 (11.6) | 99 (88.4) | 0.59 (0.23–1.47) | 0.256 | | |

Note: CI = 95% Confidence Interval, NA = Not applicable (not retained in the model), COR = Crude Odd Ratio, AOR = Adjusted Odd ratio.

with a good knowledge of malaria were 13 times more likely to use LLINs than households that do not have a good knowledge of malaria (OR: 13.27, 95% CI: 1.77–99.63). In comparison to poor households, those with middle economic status (OR: 2.01, 95% CI: 0.93–4.35) and wealthier households (OR: 6.07, 95% CI: 2.28–16.15) were more likely to use LLIN (Table 4).

When adjusted for other socio demographic characteristics and knowledge of malaria, ethnic group, number of household members, economic status of household and knowledge of malaria were significantly associated with the use of LLIN by nomads (Table 4). Daza respondents (AOR: 0.23, 95% CI: 0.09–0.56) were around one quarter as likely to use LLINs in compared with Arab respondents. In comparison to households with less than 6 members, those with 6–8 members (AOR: 2.20, 95% CI: 1.03–4.74), those with 9–11 members (AOR: 3.62, 95% CI: 1.44–9.05) and those with more than 11 members (AOR: 6.68, 95% CI: 3.19–14.01) were more likely to use LLIN respectively. In comparison to poor households, the middle (AOR: 2.78, 95% CI: 1.17–6.62) and rich households (AOR: 6.68, 95% CI: 3.19–14.01) were more likely to use LLIN. In addition, households that have a good knowledge of malaria (AOR: 12.77, 95% CI: 1.58–102.99) were 13 times more likely to use LLIN (Table 4).

## Discussion

This study assessed levels of malaria knowledge and factors associated with mosquito net use in the three main nomadic ethnic groups of Chad. Results show that in general nomads surveyed were aware of malaria risk and they reported the rainy season (July–September) as period of high transmission. However, depending on their position at the beginning or end of rainy season, their perception of risk varied. For example, Daza respondents who mostly live in areas where rainy season starts lately and ends earlier, perceived less risk of malaria in June

**Table 4. Factors associated with the use of LLIN among nomads.**

| Variables | Categories | Poor n = 187 | Accurate n = 87 | COR (95% CI) | p–value | AOR (95% CI) | p–value |
|---|---|---|---|---|---|---|---|
| **Ethnic group** | Arab (ref) | 62 (59.6) | 42 (40.4) | 1 | | 1 | |
| | Daza | 73 (90.1) | 8 (9.9) | 0.16 (0.07–0.37) | < 0.001 | 0.23 (0.09–0.56) | 0.001 |
| | Fulani | 52 (58.4) | 37 (41.6) | 1.05 (0.59–1.87) | 0.867 | 1.09 (0.56–2.10) | 0.801 |
| **Size of household** | 2–5 (ref) | 58 (77.3) | 17 (22.7) | 1 | | 1 | |
| | 6–8 | 86 (71.7) | 34 (28.3) | 1.35 (0.69–2.64) | 0.382 | 2.20 (1.03–4.74) | 0.043 |
| | 9–11 | 34 (63.0) | 20 (37.0) | 2.01 (0.93–4.35) | 0.077 | 3.62 (1.44–9.05) | 0.006 |
| | $\geq$12 | 9 (36.0) | 16 (64.0) | 6.07 (2.28–16.15) | < 0.001 | 10.87 (3.31–35.72) | < 0.001 |
| **Wealth categories** | Poorest (ref) | 92 (83.6) | 18 (16.4) | 1 | | 1 | |
| | Middle | 38 (71.7) | 15 (28.3) | 2.02 (0.92–4.41) | 0.079 | 2.78 (1.17–6.62) | 0.021 |
| | Richest | 57 (51.4) | 54 (48.6) | 4.84 (2.59–9.07) | < 0.001 | 6.68 (3.19–14.01) | < 0.001 |
| **Knowledge of malaria** | Poor (ref) | 25 (96.2) | 1 (3.8) | 1 | | 1 | |
| | Accurate | 162 (65.3) | 86 (34.7) | 13.27 (1.77–99.63) | 0.012 | 12.77 (1.58–102.99) | 0.017 |
| **Gender** | Female (ref) | 58 (32.0) | 30 (10.1) | 1 | | NA | NA |
| | Male | 129 (68.0) | 57 (9.0) | 0.85 (0.50–1.47) | 0.567 | | |
| **LLIN mentioned as preventive method** | No | 53 (94.6) | 3 (5.4) | 1 | | NA | NA |
| | Yes | 134 (61.5) | 84 (38.5) | 11.07 (3.35–36.58) | < 0.001 | | |
| **Condition of LLIN owned** | Bad (ref) | 34 (65.4) | 18 (34.6) | 1 | | NA | NA |
| | Good | 132 (65.7) | 69 (34.3) | 0.99 (0.52–1.87) | 0.969 | | |
| **Head of household status** | Widowed/ divorced (ref) | 21 (100.0) | 0 (0.0) | 1 | | NA | NA |
| | Married monogamy | 144 (69.9) | 62 (30.1) | 0.37 (0.20–0.72) | 0.003 | | |
| | Married polygamy | 22 (46.8) | 25 (53.2) | 1 | | | |
| **Number children under 5 years** | 1–2 (ref) | 144 (72.7) | 54 (27.3) | 1 | | NA | NA |
| | $\geq$3 | 43(56.6) | 33 (43.4) | 2.05 (1.18–3.55) | 0.011 | | |
| **LLIN owned** | No (ref) | 185 (100.0) | 0 (0.0) | NA | NA | NA | NA |
| | Yes | 1 (1.1) | 87 (98.9) | | | | |

NA = Not applicable (not retained in the model).

and October compared with Arab and Fulani who leave the Sudanian zone later in the year. However, more investigation is required to ensure that nomads are not perceived malaria as a seasonal disease which can be a limit to the control of the disease.

Good knowledge of period of high transmission in the rainy season may lead nomads to attribute malaria cases to environmental factors such as water and heat/sun. Regarding the heat or sun, its association to malaria cause may be due to the intermittent fever of malaria manifestations as stated in others studies [51]. Regarding spiritual influences on perception of malaria, few nomads surveyed associated malaria with religious or supernatural forces which is the contrary to the results from others studies [51]. This situation may be due to community experiences of malaria case management since it is the most common reason for seeking health care in health facilities [1].

In contrast to knowledge of causes of malaria, participants had a good knowledge of its symptoms. Around nine out of ten nomads surveyed (90.67%) had a good knowledge of malaria with score above 2.5. The main factor associated with knowledge of malaria was the ethnic group with Fulani respondents more likely to have a good knowledge of malaria. An informal discussion with an officer in charge of vaccination of nomads suggest that among nomad ethnic groups, the Fulani group is that frequently seek for health facilities and that adhere more to immunization interventions. Thus, this nomadic group may benefit from health communication at health facilities than others groups.

In addition, the ownership of LLIN was associated to the good knowledge of malaria. This result can be explained by the effect of communication during the distribution of LLIN as most of LLINs may be received in the health center by pregnant women during their first Antenatal Consultation (ANC) or by children from 0–11 months during routine vaccination.

Most nomadic households surveyed owned at least one mosquito net (91.4%), although less than half of households surveyed (32.0%) owned at least one LLIN even though the survey happened 1–2 months after the LLIN mass distribution campaign. The coverage of LLINs among nomads was low in comparison to others studies in the general population in Chad, for which coverage has been estimated at 67.1%– 73.4% [2, 46]. The low coverage of LLINs among nomads can be explained by the lack of availability of that type of mosquito in local markets; most purchase mosquito nets themselves due to the fact that their communities are not included in census enumerations of the LLIN campaigns. In addition, most of them may miss the opportunity to receive LLINs from the routine distribution because of their frequent travel in remote areas which reduce their contact with the health system [25]. This result highlight the necessity to improve the availability of LLINs to nomadic community as they are at higher risk of malaria in comparison to the settled population [3, 52]. In addition, the improvement of availability of LLINs should go alongside with others malaria interventions such as ITPp and SMC for which only 22.1% of nomadic women have received at least one dose during their last pregnancy and 22.7% of nomadic households have received a visit by distributors (Table 2). This is particularly important since the national policy against malaria requires specific strategies for the nomadic population [9]. However, neither the former strategic plan against malaria mentions this specifically [53], nor has the strategic plan of the National Health Program for nomads was developed.

On the contrary to the knowledge of SMC (7.9%) and ITPp (5.8%) as preventive method for malaria, the overall level of malaria knowledge and the awareness of mosquito net as preventive method for malaria were good. In addition, as highlighted by theories of health behavior change [37], both ownership of mosquito net and the knowledge have been translated into its use. These results are in concordance with findings from other studies that demonstrated the association between malaria knowledge with preventive behaviors' related to malaria in sub–Saharan Africa [27, 28]. However, more than half of nomads surveyed go under mosquito net later than the recommended time. An informal discussion with nomad respondents show that mostly they are busy with outside routine activities before 7PM: coming home with herds around 6PM, following by the extraction of milk and feeding younger animals. Generally this routine ends after 7PM. However, this finding shows the necessity to improve quality of information provides to nomadic communities on the mosquito net use. Such information should retain the prayer time of 6PM as the time for groups most at risk (pregnant women and children) to go under nets since almost all nomads are Muslim [54].

The results of the multivariate logistic regression highlights that nomadic ethnic group, number of household members, economic status of household and knowledge of malaria were the main factors associated with the use of mosquito net in nomadic settings. The greater the number of household members is the more likely for them to use a mosquito net. This result is contrary to those find in Zambia and Zimbabwe where the use of LLIN decreased as household size increased [27]. This contrast can be explained by the expected household expenditure on malaria case management; nomads may not benefit from free-of-charge healthcare provision since they frequently travel in remote areas which reduce their contact with the health system [25]. Thus to avoid expect expenditure for malaria case management, large households may feel encouraged to use LLINs than the small household who can handle this; small household may have a large financial resources per household member.

In comparison to poor households, the middle and wealthier households were more likely to use LLINs. This result is concordant with those of other studies [27, 55]. Results can be explained by the ownership of nets since 92.6% (30.2% for LLIN) and 94.6% (49.1% for LLIN) of respondents from middle and wealthier households declared owning at least one mosquito net, whereas this proportion is 87.5% (16.5% for LLIN) for poor households. Poorer households may face financial barriers to accessing mosquito net as most nomads mentioned buying mosquito net themselves. This result highlights the urgent need for NMCP to distribute LLINs to nomadic groups through teams comprising members of nomadic communities, as suggested elsewhere [56, 57].

In addition, Daza respondents were less likely to use LLIN in comparison to Arabs. This result can be explained by the fact that only 10.7% of Daza respondents own at least one LLIN in comparison to Arab (41.0%) and Fulani (41.6%). In addition, Daza respondents were the more poor (54.8%) than other respondents (32.4% Arab and 36.0% Fulani). This finding highlighted that the Daza nomadic group is most urgently in need to be targeted by LLIN mass distribution campaigns for the purpose of public health and equity of access to health interventions.

Although malaria interventions should be sensible to gender, in this study as elsewhere, we did not found an association between gender and respectively knowledge of malaria [58] and the use of LLINs [27, 59]. However, the aim of this study was not to assess the sensitivity of malaria interventions to gender, thus the survey sampling was not powered to integrate the gender aspect. Therefore, further studies may explore how gender norms and roles may influence the optimal design of policies and interventions aimed at improving nomadic communities' malaria prevention practices. In the meantime, regarding the potential for women to influence positively the practice of malaria preventions [60, 61], it remains important to improve nomad women' awareness of malaria and to tailor malaria prevention considering gender.

## Strengths and limitations of the study

In this study, there could be a potential bias in measuring the LLINs use among the entire members of the respondent's household as just one member per household responds on behalf of the household. Another limitation of this study was that its assessment of good malaria prevention practices of focused on use of LLINs, to the potential exclusion of other relevant aspects of malaria prevention. In addition, the study relied on self-reported information. Thus to reduce the bias, surveyors asked to verify the presence of mosquito nets in each household. The study also relied on a cross-sectional survey conducted at the end of rainy season and the dry season when mosquito density and malaria transmission may be lower than in the rainy season. However the study can be useful for understanding factors associated with increased the likelihood of use mosquito net and testing theories of behavior change in the nomadic settings.

## Conclusions

This study revealed good overall knowledge of malaria among nomadic communities, although there is a need to improve awareness of Plasmodium falciparum-carrying mosquitoes as the vector for malaria transmission. The rate of LLINs ownership was moderate, and only a small proportion of respondents slept under LLINs during the previous night before the survey. Thus, in comparison to the settled population, nomads are at higher risk of mosquito bite and hence acquiring malaria infection. This study also revealed that the main factors associated with use of mosquito nets included the nomadic ethnic groups, the number of household

members, the economic status of household and the knowledge of malaria. Further progress in malaria prevention can be achieved by improving nomads' access to LLIN mass distribution campaigns, and the quality of health information provided to nomad communities; such information can be delivered through the nomads community networks. This study can inform the National Health Program for nomads to design policies to improve nomadic communities' knowledge of malaria prevention and promoting LLIN use as requested by the national policy against malaria.

## Supporting information

**S1 File. Wealth index calculation.**
(DOCX)

**S2 File. Form questionnaire.**
(XLSX)

**S3 File. Database paper STATA.**
(DTA)

**S4 File. Inclusivity in global research.**
(DOCX)

## Acknowledgments

The authors express their gratitude to the study participants who shared their views and experiences on malaria and preventive interventions. We acknowledge contributions of Kabo Karadjom, Bianzoumbé Jonas and Issa Younous for data collection.

## Author Contributions

**Conceptualization:** Azoukalné Moukénet, Kebféné Moundiné, Ngarkodje Ngarasta, Ibrahima Seck.

**Data curation:** Azoukalné Moukénet.

**Formal analysis:** Azoukalné Moukénet, Sol Richardson, Jean Laoukolé, Ngarkodje Ngarasta, Ibrahima Seck.

**Funding acquisition:** Azoukalné Moukénet.

**Investigation:** Azoukalné Moukénet, Kebféné Moundiné, Ngarkodje Ngarasta, Ibrahima Seck.

**Methodology:** Azoukalné Moukénet, Sol Richardson, Ngarkodje Ngarasta.

**Project administration:** Azoukalné Moukénet.

**Software:** Azoukalné Moukénet.

**Validation:** Sol Richardson, Kebféné Moundiné, Jean Laoukolé, Ngarkodje Ngarasta, Ibrahima Seck.

**Visualization:** Azoukalné Moukénet.

**Writing – original draft:** Azoukalné Moukénet.

**Writing – review & editing:** Azoukalné Moukénet, Sol Richardson, Kebféné Moundiné, Jean Laoukolé, Ngarkodje Ngarasta, Ibrahima Seck.

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
