## [Decision Letter · Decision Letter 0]

21 Dec 2021

PONE-D-21-38854Knowledge and practices surrounding malaria and LLIN use among Arab, Dazagada and Fulani pastoral nomads in ChadPLOS ONE

Dear Dr. Moukénet,

Thank you for submitting your manuscript to PLoS ONE. After careful consideration, we feel that your manuscript will likely be suitable for publication if the authors revise it to address specific points raised by the reviewer. According to the reviewer, there are some specific areas where further improvements would be of substantial benefit to the readers, particularly in the methods and results. Also, discussion should be revised.   For your guidance, a copy of the reviewers' comments was included below.  

We look forward to receiving your revised manuscript.

Kind regards,

Luzia Helena Carvalho, Ph.D.

Academic Editor

PLOS ONE

Journal Requirements:

When submitting your revision, we need you to address these additional requirements. 1. Please ensure that your manuscript meets PLOS ONE's style requirements, including those for file naming. The PLOS ONE style templates can be found at https://journals.plos.org/plosone/s/file?id=wjVg/PLOSOne_formatting_sample_main_body.pdf and https://journals.plos.org/plosone/s/file?id=ba62/PLOSOne_formatting_sample_title_authors_affiliations.pdf
 2. Please include a complete copy of PLOS’ questionnaire on inclusivity in global research in your revised manuscript. Our policy for research in this area aims to improve transparency in the reporting of research performed outside of researchers’ own country or community. The policy applies to researchers who have travelled to a different country to conduct research, research with Indigenous populations or their lands, and research on cultural artefacts. The questionnaire can also be requested at the journal’s discretion for any other submissions, even if these conditions are not met.  Please find more information on the policy and a link to download a blank copy of the questionnaire here: https://journals.plos.org/plosone/s/best-practices-in-research-reporting. Please upload a completed version of your questionnaire as Supporting Information when you resubmit your manuscript.

Reviewers' comments:

Reviewer's Responses to Questions

**Comments to the Author**

1. Is the manuscript technically sound, and do the data support the conclusions?

Reviewer #1: Yes

2. Has the statistical analysis been performed appropriately and rigorously? 

Reviewer #1: Yes

3. Have the authors made all data underlying the findings in their manuscript fully available?

Reviewer #1: Yes

4. Is the manuscript presented in an intelligible fashion and written in standard English?

Reviewer #1: Yes

5. Review Comments to the Author

Reviewer #1: Dear authors, congratulations for the work done. Your manuscript is well-written and carefully explains the results and conclusions of an interesting, well-designed research. The scientific background is well explained, you have chosen a correct study design and you clearly describe the participants, variables, data sources and statistical methods.

I only have some minor comments for your consideration:

- Data collection method: missing information about how the questionnaire was delivered. Was it auto-filled by the respondent or was it deliver and filled by a health worker?

- Line 332-334: the authors explain that there is a necessity to improve quality of information because the nomad population go under the mosquito net late than 7pm. Can you only explain these findings because the quality of information provided is poor or do you think the population are after 7pm still doing important daily activities outdoor (work, cultural or housing-related)? Do you think this behaviour will improve with better information?

- Line 335: the authors highlight the importance of the education on malaria knowledge for women (which will increase use of LLIN in the household). Many studies have highlighted this finding; however, it would be nice to have results on this relationship within your data. I suggest the authors to analyse this relationship in their data (and include it in Table 3 and Table 4) so it will support their recommendation.

- The authors highlight the importance of nomadic population been targeted in LLIN mass campaigns. Most of them (91.4%) had at least 1 mosquito net in their household (not LLIN) and most of them (87.1%) have slept under the net the previous night. If LLINs are given to this population, it seems they will use them as they use the other nets. I agree that improving access to LLIN in the nomadic population is really important; however, in the Discussion, I would also focus the attention on the other strategies that these nomadic groups seem are not receiving: access to SMC, ANC services and IPT. Have you collected information on these strategies in your survey?

6. PLOS authors have the option to publish the peer review history of their article (what does this mean?). If published, this will include your full peer review and any attached files.

Reviewer #1: **Yes: **Pere Millat-Martínez

---

## [Author Response · Author response to Decision Letter 0]

3 Jan 2022

5. Review Comments to the Author

Reviewer #1: Dear authors, congratulations for the work done. Your manuscript is well-written and carefully explains the results and conclusions of an interesting, well-designed research. The scientific background is well explained, you have chosen a correct study design and you clearly describe the participants, variables, data sources and statistical methods.

I only have some minor comments for your consideration:

Thank you

- Data collection method: missing information about how the questionnaire was delivered. Was it auto-filled by the respondent or was it deliver and filled by a health worker?

Thank you for highlighted this. We have integrated your comment into the Data collection method section accordingly Line 142 – 144 “The survey questionnaire was administered in February and October 2021 by three trained data collectors fluent in the local languages and used to collect data for coverage of nomad’s children immunization.”

- Line 332-334: the authors explain that there is a necessity to improve quality of information because the nomad population go under the mosquito net late than 7pm. Can you only explain these findings because the quality of information provided is poor or do you think the population are after 7pm still doing important daily activities outdoor (work, cultural or housing-related)? Do you think this behaviour will improve with better information?

Thank you. You raised a good point here and we have integrated the comment in line 334 – 343 “An informal discussion with nomad show that mostly they are busy with outside routine activities before 7pm: coming home with herds around 6pm, following by the extraction of milk and cattle feeding their kids. Generally this routine ends after 7pm. However, this finding shows the necessity to improve quality of information provides to nomadic communities on the mosquito net use. Such information should retain the prayer time of 6pm as the time for group most at risk (pregnant women and children) to go under net since almost all nomads are Muslim (55)”. 

- Line 335: the authors highlight the importance of the education on malaria knowledge for women (which will increase use of LLIN in the household). Many studies have highlighted this finding; however, it would be nice to have results on this relationship within your data. I suggest the authors to analyse this relationship in their data (and include it in Table 3 and Table 4) so it will support their recommendation.

Thank you to mention this. We have run analysis and the gender was nor associated to knowledge, neither to the practice (included in Table 3 and Table 4). We have integrated this finding accordingly to the manuscript and deleted former recommendation, line 339 – 342 “Such information should target women as it has proven in other studies that, increased women’s knowledge of malaria can improve net use by individuals and/or members of their household (54). retain the prayer time of 6pm as the time for group most at risk (pregnant women and children) to go under net since almost all nomads are Muslim (55).”

- The authors highlight the importance of nomadic population been targeted in LLIN mass campaigns. Most of them (91.4%) had at least 1 mosquito net in their household (not LLIN) and most of them (87.1%) have slept under the net the previous night. If LLINs are given to this population, it seems they will use them as they use the other nets. I agree that improving access to LLIN in the nomadic population is really important; however, in the Discussion, I would also focus the attention on the other strategies that these nomadic groups seem are not receiving: access to SMC, ANC services and IPT. Have you collected information on these strategies in your survey?

Thank you for highlighted this comment. Although the objective of our study this time was to focus on LLINs, we have integrated your comment in Table 2 (“SMC mentioned as preventive method”, “ITP mentioned as preventive method”, “Received visit for SMC” and “Received at least one dose IPT during last pregnancy”) and in the discussion section line 325 - 327 “In addition, the improvement of availability of LLINs should go alongside with others malaria interventions such as ITP and SMC for which only 22.1% of nomadic women have received at least one dose during their last pregnancy and 22.7% of nomadic households have received a visit of distributors (Table 2).” And line 331 “On the contrary to the knowledge of SMC (7.9%) and ITP (5.8) as preventive method for malaria, the overall level of malaria knowledge and the awareness of mosquito net as preventive method for malaria were good.”

---

## [Decision Letter · Decision Letter 1]

23 Feb 2022

PONE-D-21-38854R1Knowledge and practices surrounding malaria and LLIN use among Arab, Dazagada and Fulani pastoral nomads in ChadPLOS ONE

Dear Dr. Azoukalné Moukénet,

Thank you for submitting your manuscript to PLOS ONE. After careful consideration, we feel that it has merit but does not fully meet PLOS ONE’s publication criteria as it currently stands. Therefore, we invite you to submit a revised version of the manuscript that addresses the points raised during the review process.

We look forward to receiving your revised manuscript.

Kind regards,

Marti Vall, PhD, MD

Academic Editor

PLOS ONE

Journal Requirements:

Additional Editor Comments (if provided):

This is an interesting article that provides information about practices related with LLIN and knowledge about malaria among three pastoral Chadian nomad groups. It may inform Chadian programmes for nomads to design preventive policies for this population.

Minor comments:

1) Define abbreviations when they first appear in text

2) Refine sentence in Line 62 since reference #11 refers to The Gambia not to Chad

3) Line 138. Specify who collected the data

4) Line 181. Refine and justify definition of good malaria practices (both conditions: to own and to sleep?). Is it enough to consider good practice to sleep under LLIN just the night before the survey?

5) Line 210. 68% respondents were male. Discuss the gender implications of the results of the survey. Would it had been different if 68% of respondents were female? (Ref # 54)

6) Lines 262-266. Correct interpretation of 6 times (0.16) and 3 times (0.37) of OR

7) Line 294. Refine sentence, since 77% of nomads know malaria season

8) Line 374. Refine sentence about meaning of "root causes of malaria"

Reviewers' comments:

Reviewer's Responses to Questions

**Comments to the Author**

1. If the authors have adequately addressed your comments raised in a previous round of review and you feel that this manuscript is now acceptable for publication, you may indicate that here to bypass the “Comments to the Author” section, enter your conflict of interest statement in the “Confidential to Editor” section, and submit your "Accept" recommendation.

Reviewer #1: All comments have been addressed

2. Is the manuscript technically sound, and do the data support the conclusions?

Reviewer #1: Yes

3. Has the statistical analysis been performed appropriately and rigorously? 

Reviewer #1: Yes

4. Have the authors made all data underlying the findings in their manuscript fully available?

Reviewer #1: Yes

5. Is the manuscript presented in an intelligible fashion and written in standard English?

Reviewer #1: Yes

6. Review Comments to the Author

Reviewer #1: (No Response)

7. PLOS authors have the option to publish the peer review history of their article (what does this mean?). If published, this will include your full peer review and any attached files.

Reviewer #1: **Yes: **Pere Millat-Martínez

---

## [Author Response · Author response to Decision Letter 1]

16 Mar 2022

Response to Reviewers

Comments to the Author

Additional Editor Comments (if provided):

This is an interesting article that provides information about practices related with LLIN and knowledge about malaria among three pastoral Chadian nomad groups. It may inform Chadian programmes for nomads to design preventive policies for this population.

Minor comments:

1) Define abbreviations when they first appear in text

Thanks for this comment. The manuscript has been reviewed accordingly.

2) Refine sentence in Line 62 since reference #11 refers to The Gambia not to Chad

Thank you. The sentence has been refine into “Among malaria prevention strategies adopted in Chad, intermittent preventive treatment have been shown to be effective in preventing malaria among pregnant women elsewhere (11)”.

3) Line 138. Specify who collected the data

Thank you for this comment. Line 138 is related to sampling and we did not specify who collected data in that section. However, in section “Data collection method” we have specified in line 155-163 “The survey questionnaire was administered in February and October 2021 by three trained data collectors fluent in the local languages and used to collect data for nomad immunization programs”.

4) Line 181. Refine and justify definition of good malaria practices (both conditions: to own and to sleep?). Is it enough to consider good practice to sleep under LLIN just the night before the survey?

Thank you for raising this point. We have revised the sentence as following “Good malaria prevention practices were assessed based on the ownership and use of LLINs at night”. 

We agree with your comment that good malaria practices are not limited to owning and sleeping under a LLIN, and we have integrated this point as a limitation of the study in the “Strengths and limitations of the study” section, on line 384: “In addition, another limitation of this study was that its assessment of good malaria prevention practices of focused on use of LLINs, to the potential exclusion of other relevant aspects of malaria prevention.” 

5) Line 210. 68% respondents were male. Discuss the gender implications of the results of the survey. Would it had been different if 68% of respondents were female? (Ref # 54)

Thank you for bringing up this issue, which we agree represents a ‘blind spot’ for our study. We have discussed the gender aspect in lines 390 – 398 “Although malaria interventions should be sensible to gender, in this study as elsewhere, we did not found an association between gender and respectively knowledge of malaria (58) and the use of LLINs (27,59). However, the aim of this study was not to assess the sensitivity of malaria interventions to gender, thus the survey sampling was not powered to integrate the gender aspect. Therefore, further studies may explore how gender norms and roles may influence the optimal design of policies and interventions aimed at improving nomadic communities’ malaria prevention practices. In the meantime, regarding the potential for women to influence positively the practice of malaria preventions (60,61), it remain important to improve nomad women’ awareness of malaria and to tailor malaria prevention considering gender”.

6) Lines 262-266. Correct interpretation of 6 times (0.16) and 3 times (0.37) of OR

We have changed into “one-sixth times more likely …” and “around one-third times more likely…” 

7) Line 294. Refine sentence, since 77% of nomads know malaria season

Yes, but the percentage 90.64% mentioned is related to those have knowledge score above 2.5 (see Knowledge of malaria line 177). The definition of knowledge is not just knowing the correct season.

8) Line 374. Refine sentence about meaning of "root causes of malaria"

Thank you, we have modified the sentence, which now calls for the need to improve “malaria awareness of Plasmodium falciparum-carrying mosquitoes as the vector for malaria transmission” among nomadic populations.

---

## [Editor Report · Decision Letter 2]

22 Mar 2022

PONE-D-21-38854R2Knowledge and practices surrounding malaria and LLIN use among Arab, Dazagada and Fulani pastoral nomads in ChadPLOS ONE

Dear Dr. Moukénet,

Thank you for submitting your manuscript to PLOS ONE. After careful consideration, we feel that it has merit but does not fully meet PLOS ONE’s publication criteria as it currently stands. Therefore, we invite you to submit a revised version of the manuscript that addresses the points raised during the review process.

We look forward to receiving your revised manuscript.

Kind regards,

Marti Vall, PhD, MD

Academic Editor

PLOS ONE 

Journal Requirements:

Additional Editor Comments:

The authors have addressed the comments of the reviewers. However, they have responded to them with two different revisions of the manuscript. They have to incorporate both comments in a single final version.

---

## [Author Response · Author response to Decision Letter 2]

24 Mar 2022

Thanks. We did not know that all comments from the beginning should be incorporate into the single original manuscript. We have corrected the manuscript accordingly. 

We incorporated also all responses to reviewers into one single (current) file.

---

## [Editor Report · Decision Letter 3]

30 Mar 2022

Knowledge and practices surrounding malaria and LLIN use among Arab, Dazagada and Fulani pastoral nomads in Chad

PONE-D-21-38854R3

Dear Dr. Moukénet,

We’re pleased to inform you that your manuscript has been judged scientifically suitable for publication and will be formally accepted for publication once it meets all outstanding technical requirements.

Kind regards,

Marti Vall, PhD, MD

Academic Editor

PLOS ONE

Additional Editor Comments (optional):

The manuscript has been improved and is acceptable for publication.
---

## [Editor Report · Acceptance letter]

6 Apr 2022

PONE-D-21-38854R3 

Knowledge and practices surrounding malaria and LLIN use among Arab, Dazagada and Fulani pastoral nomads in Chad 

Dear Dr. Moukénet:

I'm pleased to inform you that your manuscript has been deemed suitable for publication in PLOS ONE. Congratulations! Your manuscript is now with our production department. 

Kind regards, 

on behalf of

Dr. Marti Vall 

Academic Editor

PLOS ONE